# NEXT-SCALE AUTOREGRESSIVE MODELS ARE ZERO-SHOT SINGLE-IMAGE OBJECT VIEW SYNTHESIZERS

## ABSTRACT

Learning to synthesize novel views without explicit 3D representations or hand-crafted 3D inductive bias has recently gained attention: it is simpler, more formally direct, and better aligned with the lesson that scalable learning paradigms with less assumptions built into architectural design (*e.g.*, regarding geometry) often win. However, the current dominant solutions are diffusion-based, which typically suffer from problems like slow inference. We introduce ArchonView, the first autoregressive model for zero-shot single-image, object-centric novel view synthesis (NVS), achieving substantially faster inference, higher accuracies, and notably not relying on fine-tuning of 2D generative checkpoints (challenging the common assumption that 2D priors are required in diffusion-based NVS). We design innovative methods of both global and local conditioning to suit characteristics of the NVS task. Crucially, a naïve application of next-scale autoregression fails; we identify two design choices that unlock performance: local conditioning pre-filling, and removing global AdaLN at the classifier head. ArchonView delivers state-of-the-art zero-shot results across six standard benchmarks (GSO, ABO, OmniObject3D, RTMV, NeRF-Synthetic, ShapeNet), while being several times faster than diffusion baselines (*e.g.*, 0.22s *v.s.* 1.7–1.8s per view at matched parameter count). It consistently improves synthesis accuracy, and scales predictably with both model size (135M–2B) and data size, exhibiting clear scaling-law-like trends. Our findings suggest a paradigm shift and challenge an existing assumption: first, for object-centric NVS, next-scale autoregression can be faster, simpler, and more accurate than diffusion; and second, priors obtained from fine-tuning 2D-pretrained models may not be necessary for generative NVS. Our code is open-sourced at https://anonymous.4open.science/r/ArchonView/.

## 1 INTRODUCTION

Humans, living in a 3D world, naturally infer the complete 3D structure of objects from a single 2D view, leveraging prior knowledge and spatial reasoning. If machines could achieve the same, particularly in a zero-shot manner for unseen objects, it would greatly benefit fields such as 3D content creation, simulation, and real-world perception systems. Consequently, zero-shot novel view synthesis (NVS) from single object-centric images emerges as a fundamental challenge in computer vision. Since this is a highly under-constrained problem, it is typically formulated as a generative task conditioned on the input image and relative camera pose. The prevailing approach fine-tunes a 2D diffusion model to exploit implicit geometric priors learned from large-scale image datasets. Whilst this paradigm has shown promising results, it comes with several limitations.

A critical limitation of diffusion models is their inherent trade-off between speed and quality. Due to the need for multiple denoising steps through a U-Net structure, achieving high-quality outputs inevitably results in relatively slow inference. Consequently, if diffusion models were scalable, larger models would further increase inference time and computational cost, potentially rendering them impractical for real-world deployment. Furthermore, notably, to the best of our knowledge, no prior work has demonstrated scaling trends with respect to model size for single-image object-centric NVS. We consider this to perhaps be attributable to a previous common assumption, introduced in works like Liu et al. (2023), which conclude that 2D priors obtained from fine-tuning pretrained 2D generative models (such as Stable Diffusion) are necessary for zero-shot NVS. Since it is impossible for most researchers to train a 2D generative model from scratch, this has limited most current

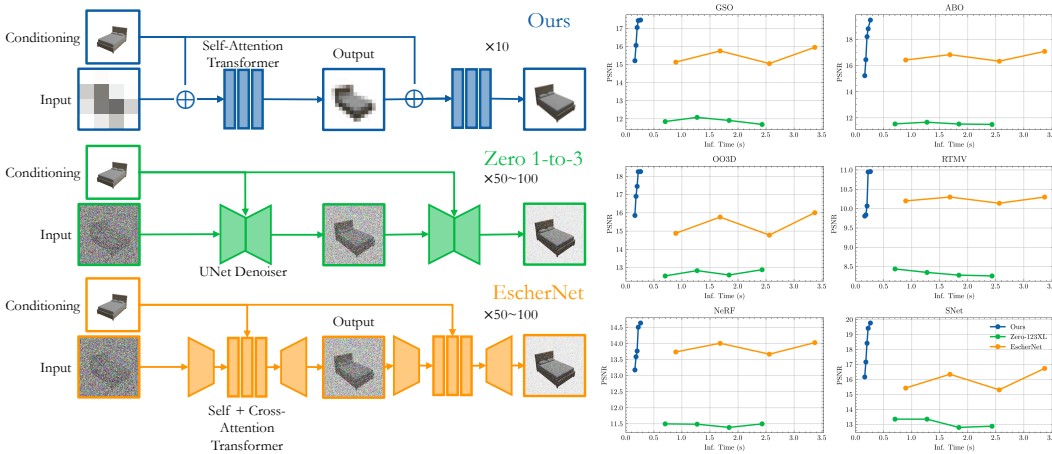

Figure 1: **Fast, accurate, and scalable novel view synthesis without fine-tuning.** We achieve better results via paradigm shift, by using an autoregressive scheme to replace previous diffusion schemes. The left shows the abstract architecture of each autoregression/diffusion step. On the right are time *v.s.* PSNR tradeoff plots, where our models scaled to different parameter sizes are compared against diffusion models with different denoising steps.

models to be based on fine-tuning Stable Diffusion, and hence remaining at the size of around 1.0B parameters.

We explore the potential for a paradigm shift: a backbone architecture which can be readily scaled up, does not require 2D pretraining, is more efficient, and outputs better results. To this end we propose Autoregression Conditioned by View (ArchonView), the first NVS model based on visual autoregressive generation. We base our method on the recently proposed next-scale autoregression backbone, which replaces raster-scan-ordered next-token prediction in conventional visual autoregression with autoregressively predicting the next resolution scale of tokens in a coarse-to-fine manner (described in Sec. 3.2). Whilst previous works have verified its scalability and applicability to many tasks in vision, so far, its applicability to NVS remains unknown.

In order to adapt autoregression to NVS, we condition the model in two ways: globally, through a posed CLIP encoding, which allows the model to gain semantic information augmented by the desired relative pose (described in Sec. 3.3); and locally, encoded by the multiscale VQVAE representation (which is also used for quantization in the main generative architecture), enforcing consistency in local details with the input image (described in Sec. 3.4). Experimentation shows that our method achieves state-of-the-art performance consistently and robustly across multiple benchmarks, scales with both model size and dataset size, and is several times faster than current diffusion-based methods. Some of our results across different evaluation datasets and comparisons with current methods are shown in Fig. 1.

In summary, our key technical contributions to the field of zero-shot single-image object NVS are as follows:

- We present a method that consistently achieves state-of-the-art performance and is also several times faster in terms of inference time compared to previous works.
- Our method does not require fine-tuning on a 2D generative model, and hence challenges the existing assumption that pretrained 2D priors are necessary for zero-shot capability in generative NVS.
- We demonstrate that our method scales with both model size and dataset size.
- We are the first to base a model on an autoregressive backbone for this task, demonstrating the potentials of next-scale autoregression.

## 2 RELATED WORKS

### 2.1 GENERATIVE MODELING

Autoregression has seen many applications to generation of language (Brown et al., 2020; Radford et al., 2018; 2019; Touvron et al., 2023), world models (Bruce et al., 2024; Lu et al., 2024; Tu

et al., 2025), videos (Kondratyuk et al., 2024; Wu et al., 2022; Yan et al., 2021), and multimodal outputs (Chameleon Team, 2024; Kelly et al., 2024; OpenAI, 2023; Sun et al., 2023). Besides being efficient and accurate enough for operational use, it has also been shown to be scalable in many tasks (Henighan et al., 2020; Kaplan et al., 2020). However, the current predominant paradigm of 2D generative modeling is indubitably diffusion, thanks to multiple groundbreaking innovations in this field (Ho & Salimans, 2021; Peebles & Xie, 2023; Rombach et al., 2022; Zhang et al., 2023). Meanwhile, though conventional visual autoregression (using a raster-scan traversal of fixed-size patches as tokens) has produced innovative techniques (Esser et al., 2021; Lee et al., 2022; Parmar et al., 2018; Van Den Oord et al., 2017; Yu et al., 2022a) and achieved some milestones (Chen et al., 2020; Ramesh et al., 2021; Razavi et al., 2019; Yu et al., 2022b), advances in diffusion left it largely irrelevant.

Recently, the newly proposed next-scale prediction paradigm of autoregression (Tian et al., 2024), replacing the conventional raster-scan next-token paradigm, has been proven effective, developed upon (Gu et al., 2024; Ren et al., 2024a;b; Tang et al., 2025), and adapted (Han et al., 2024; Li et al., 2024; 2025; Ma et al., 2024; Yao et al., 2024). Empirical evidence demonstrates its reliability in achieving accurate, efficient, and scalable results, exceeding diffusion models in many tasks where raster-scan autoregression struggles. This has reignited interest in using autoregressive models for visual generative tasks where diffusion models currently dominate.

## 2.2 Novel View Synthesis

Before the advent of generative NVS models, predominant methodologies have used implicit representations (Barron et al., 2021; 2022; 2023; Mildenhall et al., 2021), voxel-like representations (Chen et al., 2022; Fridovich-Keil et al., 2022; Sun et al., 2022; Yu et al., 2021a), explicit primitives (Huang et al., 2024; Kerbl et al., 2023; Mai et al., 2024; Müller et al., 2022), *etc.* to model 3D scenes or objects based on given views and poses, thus achieving NVS. However, in the case of sparse inputs, where few views or only one view is available, none of those models are able to produce accurate results due to the scene being severely underconstrained.

While some efforts have been made to adapt conventional frameworks to sparse-input scenarios (Chen et al., 2021; Chibane et al., 2021; Jain et al., 2021; Niemeyer et al., 2022; Xu et al., 2022; Yu et al., 2021b), their capabilities were limited, or often depended on strict hypotheses regarding geometrical priors. In addition, most aforementioned methods require specific training or fine-tuning on the scene or object in question, or were fine-tuned on a class of objects (*e.g.*, from ShapeNet (Chang et al., 2015)) and only perform well for in-distribution inputs. Thus, none achieved capability for zero-shot single-image NVS with such paradigms.

## 2.3 Generative NVS

Early works have used GANs (Goodfellow et al., 2014) as backbones for conducting NVS generatively, reframing the problem as modeling the distribution of scene views conditioned by camera pose (Chan et al., 2021; 2022; Gadelha et al., 2017; Nguyen-Phuoc et al., 2019; Niemeyer & Geiger, 2021; Schwarz et al., 2022). In late 2022 through 2023 there was a surge of works using diffusion models for NVS, often coupled with the then-recent NVS representations (*e.g.* NeRF); in particular, some works used diffusion models as priors for supervising the training of a 3D representation model (Bautista et al., 2022; Deng et al., 2023; Melas-Kyriazi et al., 2023; Sargent et al., 2024; Wang et al., 2023; Wu et al., 2024), while others directly used diffusion models with camera pose conditioning as an NVS representation backbone (Chan et al., 2023; Liu et al., 2023; Watson et al., 2023) (interestingly the latter's methodology of view generation based on latents coincides with previous works using transformers (Kulhánek et al., 2022; Rombach et al., 2021; Sajjadi et al., 2022)).

A particularly important work in this period was Zero 1-to-3 (Liu et al., 2023), which was fine-tuned on an image variation model (Pinkney, 2023) which, in turn, was tuned on Stable Diffusion (Rombach et al., 2022). Its main conclusion was that 2D diffusion models already contain 3D-aware priors, and that such priors can be directly extracted by fine-tuning pretrained 2D diffusion checkpoints. This paradigm of fine-tuning 2D diffusion models for priors spawned a line of works (Kong et al., 2024; Liu et al., 2024; Shi et al., 2023; Watson et al., 2024; Ye et al., 2024; Zheng & Vedaldi, 2024) which similarly attempt to extract 3D-aware priors from pretrained 2D diffusion models.

**Figure 2: The overall architecture for the training of ArchonView.** The predictions are classifier logit predictions, and can be converted to images after sampling based on the logit probabilities. The loss calculation is directly based on the logits and does not involve sampling.

## 2.4 OUR CONTRIBUTION

In our work, we demonstrate that autoregressive models can directly possess 3D-awareness without relying on checkpoints from 2D pretraining, differing from the conclusion of Zero 1-to-3. We also show that the autoregressive paradigm can be superior to its diffusion-based counterpart in terms of both speed and accuracy. We note that some very recent concurrent works (Kong et al., 2025; Nair et al., 2025) have also noticed the potential of transformer architectures in the task of NVS in general, but still rely on pretrained diffusion models as backbone for generation. We are the first work to show that in fact, autoregression is all one needs, and that neither diffusion nor fine-tuning are necessary. On top of this, our model achieves a very high increase in accuracy and efficiency compared to diffusion, suggesting that next-scale autoregression may be better suited to the task of zero-shot single-image NVS. The natural scalability of our model also provides it with high potential for applicational use.

We also note that there exists an adjacent line of works which directly produce 3D models of objects from a single image (Hong et al., 2024; Tang et al., 2024; Xiang et al., 2025; Zhang et al., 2025). However, as was pointed out by many previous works (Jin et al., 2025; Zheng & Vedaldi, 2024), models that explicitly maintain an underlying 3D representation sacrifice underlying NVS accuracy and efficiency considerably. While a lot of progress has been achieved in this direction, so far no image-to-3D model can directly achieve NVS performance better than pure NVS models.

## 3 METHODOLOGY

### 3.1 MOTIVATION

**Problem Formulation**    Formally, we wish to solve the following problem. Given a single input view $x$ of an underlying 3D object, as well as relative camera transformations $R \in \mathbb{R}^{3 \times 3}$ and $T \in \mathbb{R}^3$, we would like to create a probabilistic model $g(x, R, T)$ such that the output view

$$x^*_{R,T} \sim g(x, R, T) \tag{1}$$

follows the distribution of the transformed view from applying the relative camera transformation $(R, T)$ on $x$.

**Diffusion is Not All You Need**    The current predominant formulation of our problem is as a diffusion model. Specifically, common architectures based on the latent diffusion paradigm (Rombach et al., 2022) are made up of an image encoder-decoder pair $(\mathcal{E}, \mathcal{D})$, a U-Net denoiser $\epsilon_\theta$, and a conditioning encoder $\tau$. The latents $z \sim \mathcal{E}(x)$ are then corrupted with additive Gaussian noise at each step, forming noised latents $z_t$. Diffusion models are thus trained based on the objective

$$\min_\theta \mathbb{E}_{z \sim \mathcal{E}(x), \epsilon \sim \mathcal{N}(0,1), t} ||\epsilon - \epsilon_\theta(z_t, t, \tau(x, R, T))||_2^2. \tag{2}$$

However, we notice a key downside of this formulation; zero-shot capabilities in this line of research are dependent upon priors within 2D pretrained diffusion models. For instance, one can compare (Watson et al., 2023) with (Liu et al., 2023), which are both based on the presented model; the former was not tuned from 2D models and thus did not exhibit zero-shot abilities, while the latter was tuned from Stable Diffusion and used CLIP for conditioning encoding, thus allowing

Figure 3: **Structure of the multi-scale VQVAE.** The VQVAE converts the input image into a feature map, resizing it into different scales, and using a codebook shared between scales to compress the patches.

Figure 4: **The next-scale prediction transformer architecture.** Next-scale prediction passes local conditioning tokens and input tokens through multiple transformer blocks (with masking) which are conditioned by adaptive layer normalization.

zero-shot NVS. This naturally leads to a scalability issue: since the underlying checkpoints require large amounts of data and computation to train, it is difficult to scale existing models up. To the best of our knowledge, most or potentially all works in this direction are tuned from Stable Diffusion. Diffusion models also have other problems such as relatively slow inference speed (due to repeated denoising passes), which further limit their effectiveness.

## 3.2 BACKBONE ARCHITECTURE

We propose using next-scale autoregression as our backbone paradigm instead and modifying it to suit our task. Our overall architecture during training is shown in Fig. 2. Next-scale autoregression is based on predicting the next resolution scale of the image in a coarse-to-fine manner, and consists of two main parts: a multi-scale vector-quantized variational autoencoder (VQVAE) (Van Den Oord et al., 2017) which supports image tokenization (depicted in Fig. 3), and a transformer which autoregressively predicts the next scale of tokens (depicted in Fig. 4). More details regarding the implementation those two components can be found in App. C. Here, we emphasize our main innovations in design to adapt the paradigm specifically for our task of NVS:

**Adding Local Conditioning** While there are many tried-and-true methods regarding autoregression conditioning, we note that many of them do not fit our case. For instance, Li et al. (2025), using conventional autoregression, fuses conditioning and input tokens to achieve high efficiency and quality in generation. However, their fundamental logic does not trivially extend to novel-view synthesis: in most generative tasks (*e.g.*, sketch-to-paint, canny edge-conditioned generation, segmentation-conditioned generation), patches at the same positions in an image correspond closely to each other. However, in novel view synthesis, the correspondence between patches depends entirely on the camera pose desired, and hence this correspondence is broken. A demonstration of this phenonmenon for NVS is shown in Fig. 5.

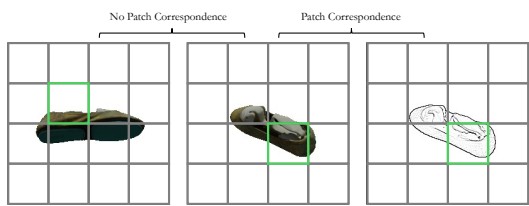

Figure 5: **Demonstration of the failure of patch correspondence.** A conditioning image and its ground-truth canny edge contain highly correlated features at patches of the same location. A conditioning image and its ground-truth novel view may have almost unrelated visual features, and even corresponding parts of the object in the image are usually in patches at different locations.

To solve this problem, we tried multiple architectures to find the most suitable conditioning for this purpose. Visualizations of their schemes are shown in Fig. 6. These include the classical prefilling, causal conditioning, and using cross-attention to fuse corresponding conditioning and input tokens of the same scale. Causal conditioning is inspired by works like Li et al. (2024), where conditioning tokens are causally masked in the same way as their corresponding input tokens of the same scale. Cross-attention (with conditioning tokens as key and value, and input tokens as key) is a variation on the idea of fusing corresponding tokens (Li et al., 2025): due to broken patch-to-patch correspon-

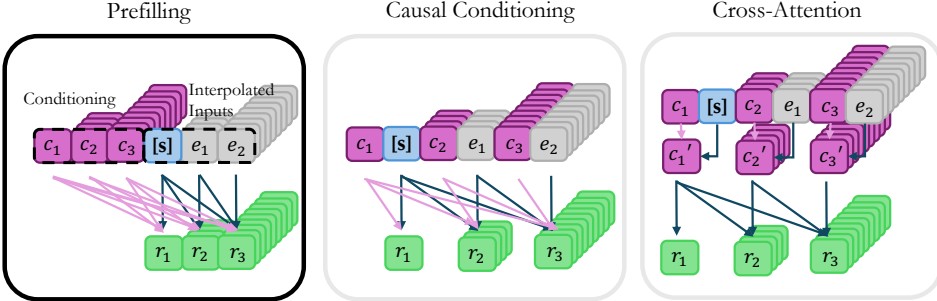

Figure 6: **Autoregressive conditioning schemes we tested.** The one that we chose (and presented in Fig. 2) was Prefilling.

dence, we instead let cross-attention automatically produce correspondences between patches of the same scale.

Results of preliminary tests show that simple prefilling turns out to exceed the other two methods by wide margins. This gives two insights into the mechanism of generative NVS. First, this means that causality between the conditioning image and the generated tokens is no longer single-direction next-scale dependency, and hence that connections from higher-resolution conditioning to lower-resolution outputs are needed. Second, this shows fusing conditional and input tokens is no longer an effective method due to correspondences being broken, even when we use more sophisticated methods than simple concatenation.

**The Devil is in the Classifier Head** We identify a seemingly innocuous and intuitive implementation feature that was not mentioned in the paper introducing next-scale autoregression (Tian et al., 2024), but turned out to be fatally detrimental to our task (as verified by our ablation studies presented in Sec. 4.2). The original implementation of the next-scale transformer applies adaptive normalization before the classifier head, which would seem as very natural and harmless, as all transformer layers also had adaptive normalization and indeed benefited from it. However, we found that in our task this significantly restricts the performance of the resulting model considerably. Removing this unleashed a significant performance improvement for our model.

We theorize that this implies global conditioning using AdaLN must be accompanied by self-attention with local conditioning tokens (as the classification logit prediction step does not involve attention transformers), and that otherwise only using global conditioning would prompt the output image to "forget" details from local conditioning. Specifically, the scale and shift for this AdaLN were derived from a linear layer, and hence are shared across all inputs, which seems irrational given the highly pose-conditioning-dependent nature of the NVS task.

### 3.3 SEMANTIC GLOBAL POSE CONDITIONING

We now start presenting the details of how we adapted the architecture via adding conditioning for our task. Firstly, we need to choose a start token that reliably captures global information from our conditioning $(x, R, T)$, because it will be used both for initializing the autoregressive procedure and for normalization of every attention layer, which means it should have complete field of view. Meanwhile, it must also be a single token because it will be mapped to the first resolution scale (which comprises of a single patch) during autoregressive inference.

To balance those two needs, we would like an encoding scheme that could condense semantic information from the image along with the query poses into a vector of size comparable to tokens. Inspired by Liu et al. (2023), we use a "Posed CLIP" embedding as follows:

$$\tau(x, R, T) = W(W_i(\text{CLIP}(x) \oplus [\theta, \sin \phi, \cos \phi, r]) + b_i) + b. \tag{3}$$

Here, $\theta$, $\phi$, and $r$ respectively stand for the relative elevation, azimuth, and radial distance of the transformation; $\oplus$ stands for concatenation along the feature dimension; $\text{CLIP}(x) \in \mathbb{R}^{768}$ is the CLIP visual embedding (Radford et al., 2021), which we use here as a global semantic encoder; and $(W_i, b_i), (W, b)$ are two pairs of linear layer parameters, where the $i$ subscript stands for identity initialization (*i.e.* $W_i$ is initialized as an identity matrix and $b$ is initialized as a zero vector).

The two layers here serve different purposes. The identity-initialized layer $W_i \in \mathbb{R}^{768 \times 772}, b_i \in \mathbb{R}^{768}$ aims to merge the relative pose information into the original CLIP features. The other layer

Table 1: **Quantitative benchmarking results.** We compare against state-of-the-art baselines using diffusion backbones on six well-established benchmarking datasets.

| | GSO | | | ABO | | | OO3D | | | |
|---|---|---|---|---|---|---|---|---|---|---|
| | PSNR (↑) | SSIM (↑) | LPIPS (↓) | PSNR (↑) | SSIM (↑) | LPIPS (↓) | PSNR (↑) | SSIM (↑) | LPIPS (↓) | Time (s) |
| Zero 1-to-3 | 13.39 | 0.7776 | 0.2672 | 12.75 | 0.7632 | 0.2901 | 13.43 | 0.7737 | 0.2723 | 1.84 |
| Zero 123-XL | 13.80 | 0.7865 | 0.2595 | 12.78 | 0.7646 | 0.2766 | 14.05 | 0.7966 | 0.2516 | 1.84 |
| EscherNet | 16.77 | 0.8275 | 0.1891 | 17.03 | 0.8381 | 0.1693 | 16.12 | 0.8294 | 0.2060 | 1.68 |
| Ours | **17.44** | **0.8491** | **0.1853** | **18.82** | **0.8725** | **0.1360** | **18.26** | **0.8622** | **0.1617** | **0.22** |

| | RTMV | | | NeRF | | | SNet | | | |
|---|---|---|---|---|---|---|---|---|---|---|
| | PSNR (↑) | SSIM (↑) | LPIPS (↓) | PSNR (↑) | SSIM (↑) | LPIPS (↓) | PSNR (↑) | SSIM (↑) | LPIPS (↓) | Params |
| Zero 1-to-3 | 8.49 | 0.5260 | 0.4772 | 10.88 | 0.6222 | 0.4146 | 13.02 | 0.7957 | 0.3288 | 1.0B |
| Zero 123-XL | 8.58 | 0.5237 | 0.4735 | 11.29 | 0.6551 | 0.3926 | 13.29 | 0.8070 | 0.3206 | 1.0B |
| EscherNet | 10.38 | 0.5327 | 0.4340 | 13.85 | 0.6783 | 0.2868 | 16.35 | 0.8450 | 0.1951 | 1.0B |
| Ours | **10.95** | **0.5739** | **0.3991** | **14.51** | **0.7025** | **0.2735** | **19.42** | **0.8927** | **0.1300** | **1.0B** |

Table 2: **Ablation study results.** We evaluate the necessity of two design choices: adding local conditioning (as in Sec. 3.4), and removing adaptive layer normalization for the classifier head.

| | GSO | | | ABO | | | OO3D | | |
|---|---|---|---|---|---|---|---|---|---|
| | PSNR (↑) | SSIM (↑) | LPIPS (↓) | PSNR (↑) | SSIM (↑) | LPIPS (↓) | PSNR (↑) | SSIM (↑) | LPIPS (↓) |
| Global Only | 13.25 | 0.7959 | 0.2672 | 13.04 | 0.7892 | 0.3116 | 15.03 | 0.8161 | 0.2523 |
| w/ Cls. Head AdaLN | 12.76 | 0.7885 | 0.2510 | 12.80 | 0.7549 | 0.2540 | 12.59 | 0.7832 | 0.2540 |
| Ours | **17.44** | **0.8491** | **0.1853** | **18.82** | **0.8725** | **0.1360** | **18.26** | **0.8622** | **0.1617** |

| | RTMV | | | NeRF | | | SNet | | |
|---|---|---|---|---|---|---|---|---|---|
| | PSNR (↑) | SSIM (↑) | LPIPS (↓) | PSNR (↑) | SSIM (↑) | LPIPS (↓) | PSNR (↑) | SSIM (↑) | LPIPS (↓) |
| Global Only | 8.74 | 0.4968 | 0.5124 | 11.80 | 0.6280 | 0.4176 | 14.52 | 0.8232 | 0.2906 |
| w/ Cls. Head AdaLN | 7.39 | 0.4771 | 0.5301 | 10.41 | 0.5958 | 0.4237 | 12.73 | 0.7929 | 0.2739 |
| Ours | **10.95** | **0.5739** | **0.3991** | **14.51** | **0.7025** | **0.2735** | **19.42** | **0.8927** | **0.1300** |

$W \in \mathbb{R}^{C \times 768}, b \in \mathbb{R}^C$, initialized normally, serves as an interface between the semantic embedding and the transformer architecture by mapping it onto a token. The $(W_i, b_i)$ layer is set to have 10 times the learning rate of other parameters, as in our experiments without this setting the gradient quickly explodes, which shows that this layer is an important information bottleneck.

We add classifier-free guidance (CFG) by randomly replacing values in the 768-dimensional posed CLIP embedding with values from a null CLIP text embedding (applying the CLIP text encoder to an empty string). This ensures that we are able to continue enjoying benefits brought by diffusion models' CFG in strengthening the impact of pose conditioning.

### 3.4 MULTI-SCALE LOCAL CONDITIONING

The global encoding is a good condition for generation because it aggregates semantic information across the entire image and has full field-of-view. However, in this process, the details of the original image are lost. Hence we need to add another source of conditioning which can directly provide the autoregressive model with portions of the input image. To this end we propose a local conditioning mechanism to provide effective conditioning information for the attention layers.

As previously shown in Fig. 2, we encode the conditioning image $x$ through the multi-scale VQVAE into tokens the same way as we encode the ground-truth $x^*$. This ensures that the conditioning tokens and tokens used during generation share the same vocabulary (from the VQVAE codebook $Z$), thus facilitating the application of self-attention in the next-scale transformer.

We extend the block triangular mask such that all conditioning tokens can affect the next attention block's conditioning and the input tokens, but are not affected by input tokens (as displayed in Fig. 4). This method of conditioning via token prepending is well-suited to the autoregressive nature of our paradigm, and does not require any architectural accommodations. We observe in our ablation studies that this greatly enhances the effectiveness of our method compared to purely using a global encoding as conditioning. We add classifier-free guidance by randomly replacing local conditioning tokens with a learnable null token.

In conventional diffusion-based methods, the predominant way of adding local conditioning is to channel-concatenate the conditioning image onto the latent noise used for diffusion. However, as previous works have noted (Shi et al., 2023), this inherently creates a false pixel-to-pixel correspondence between the conditioning image and the latent whereas the actual relationship is a lot more non-trivial (as transformations in 3D are involved). Our method, in contrast, does not inherently impose any correspondences. Since the multi-scale encoding is inherently hierarchical, not only can the transformer learn correspondence relationships without any structural assumptions, it can

Figure 7: **Qualitative results.** Visual comparisons of our method with diffusion-based prior works.

also model more complex relationships that require information from several levels of detail. This also allows for emergent aggregation of information, as our visualization of attention maps later confirms.

## 4 EXPERIMENTS

### 4.1 SETTINGS

**Training** We use the multi-scale VQVAE checkpoint from Tian et al. (2024), which was trained on the OpenImages dataset (Kuznetsova et al., 2020). We train our models on the Objaverse dataset (Deitke et al., 2023), from which we render $256 \times 256$ views with randomly sampled camera poses. The number of $16 \times 16$ patches per side for each VQVAE scale follows the progression $(1, 2, 3, 4, 5, 6, 8, 10, 13, 16)$, for a total of 10 prediction steps. The depth of the model is set to be 24 for benchmarking in order for the model size to be 1.0B, matching diffusion-based baselines. More training details are described in App. B.

**Evaluation** We conduct evaluation across six common datasets: Google Scanned Objects (GSO) (Downs et al., 2022), Amazon Berkeley Objects (ABO) (Collins et al., 2022), OmniObject3D (OO3D) (Wu et al., 2023), Ray-Traced Multi-View Synthetic (RTMV) (Tremblay et al., 2022), NeRF Synthetic (NeRF) (Mildenhall et al., 2021), and ShapeNet Core (SNet) (Chang et al., 2015). None of those datasets overlap with our training set (Objaverse), which means that all of the experimental results are zero-shot inference. To the best of our knowledge, this is the most comprehensive evaluation set (in terms of the number of independent datasets used) assembled in our line of work. Unlike previous works which, for unknown reasons, have standardized using an evaluation setting easier than the training setting, we use an evaluation setting that is the same as the training setting to test model robustness, detailed in App. A.

### 4.2 RESULTS

**Benchmarking** Quantitative results from our benchmarking are shown in Tab. 1. As shown, our model, while being of the same size as the Stable Diffusion-tuned baselines, is up to more than 8 times faster than those baselines (under their officially recommended settings), and consistently produces significantly better results across our 6 vastly distinct benchmarks.

**Qualitative Comparison** We present qualitative results in Fig. 7. As shown, our model produces more accurate reconstructions, and also can achieve more realistic and feasible results (*e.g.*, GSO shoe) even though it has seen far fewer visual information than the diffusion-based baselines (which have gone through 2D pretraining). Furthermore, our model is also good at localizing objects after transformation even in cluttered scenes (*e.g.*, RTMV), and has a good understanding of geometrical structure (*e.g.*, SNet table). Even when outputs are inaccurate due to uncertainty regarding factors such as lighting (*e.g.*, NeRF lego bulldozer), it continues to present accurate geometries and feasible lighting/texture. Surprisingly, although one might expect diffusion-based models would be more capable of predicting unseen portions of objects due to its knowledge of 2D priors, it seems that

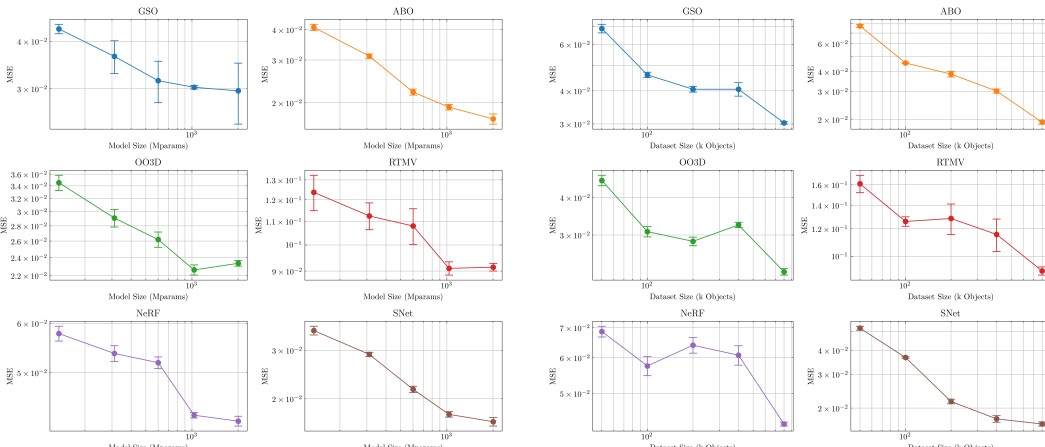

(a) **ArchonView scaling with model size.** The five tested models have depth 12, 16, 20, 24, and 30.

(b) **ArchonView scaling with dataset size.** We randomly subsample subsets from our training set.

Figure 8: **Scaling behavior we observed from Archonview.** Error bars are $\pm$ one standard deviation from five repetitions each.

ours often better (*e.g.*, ABO — note how EscherNet creates counterintuitive holes and Zero-123XL struggles to change viewpoints correctly). This supports our hypothesis that priors obtained from fine-tuning 2D generative models may not be necessary.

**Ablation** We compare our model with a version that uses only global encodings, and one that keeps AdaLN with its classifier head. The results in Tab. 2 show that both of the tested design choices were necessary for the model to achieve acceptable results. This shows that our local embedding effectively encodes local features, enabling our model to synthesize high-quality and precise novel views. In addition, this verifies our assessment on the effect of AdaLN on the classifier head, proving that local conditioning is always necessary after global conditioning to prevent loss of information.

**Scaling** We investigate the model's scaling behavior with respect to model size and dataset size (we do not consider scaling with computation due to the difficulty of rigorously defining an optimal stopping point, as in our case performance is not directly tied to token accuracy unlike in other tasks like LLMs). Results are shown in Figs. 8a and 8b. It seems that adding more data would likely continue to improve model performance considerably. As for model size, results demonstrate preliminary evidence of scaling law-like behavior below 1B parameters, and the performance of larger models is likely bottlenecked by dataset size. We expect the scaling behavior to hold when data, model size, and training compute are scaled up simultaneously, although a necessary limit will be imposed by entropy from the uncertainty of unseen parts.

**Zero-Shot Spatial Understanding** In order to demonstrate that our models has indeed acquired the ability of zero-shot spatial understanding, we visualize the attention maps of conditioning tokens in 9. The heatmap represents how much information each token in the resulting image draws from the conditioning token, and bases it on averaging the attention scores over each of the heads and layers. The results show a clear pattern: output tokens spatially or semantically related to the input token tend to have higher attention scores. Notably, a patch of the white background always corresponds to the boundary of the object, showing global geometric understanding. Tokens surrounding the input token in 2D space also tend to have higher attention scores, which is because the next-scale approach naturally clusters local information onto appropriate places in pixel space. The results support that our model has meaningful emergent capability for 3D-aware spatial understanding and information aggregation.

## 5 DISCUSSION

According to Liu et al. (2023), diffusion-based object-centric NVS models work by adapting visual priors inherent in pretrained 2D diffusion models. This, in turn, causes heavy reliance on large pretrained models such as Stable Diffusion (which was trained on over 2B images (Schuhmann et al., 2022)). In contrast, we only use the "fine-tuning" dataset of previous works (with 800k 3D objects)

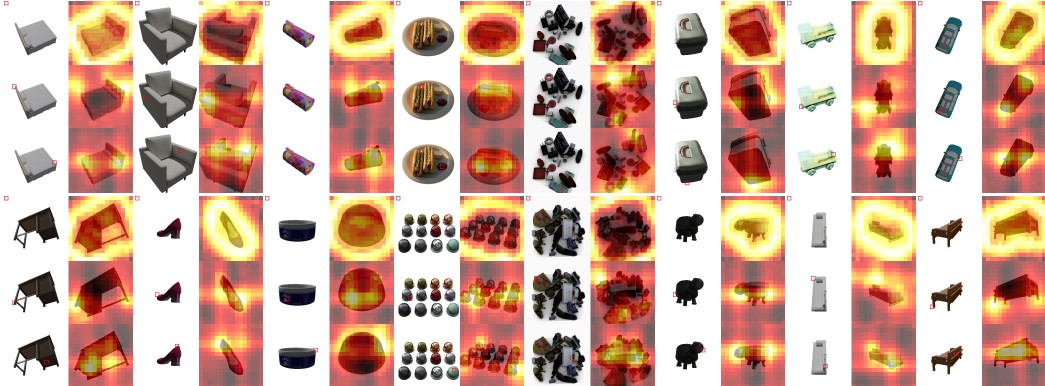

Figure 9: **Demonstration of emergent zero-shot spatial understanding capabilities.** Each of the images on the left are the conditioning images, and the image on the right is the target image for reference. The red-highlighted patch on the left corresponds to the conditioning token we choose, and the heatmap shows its attention map.

and achieved significantly superior results. This demonstrates that under the autoregressive formulation, the 3D objects already contain enough information for the model to accurately, efficiently, and scalably conduct object-centric NVS. Furthermore, trends in Fig. 8b show that increasing the dataset size beyond 800k (Objaverse) is likely to yield further positive results, which gives a clear direction for the next step of scaling ArchonView towards operational usage.

By using an autoregressive paradigm, we also avoid blurry output that defy prior knowledge of real-life objects which often occur in diffusion models, as unlike diffusion models which are continuous, the discrete representation enforced by the codebook directly filters out cases with low prior probability. This is beneficial for output fidelity and visual quality.

In addition, our superior speed and accuracy suggests the potential for a paradigm shift from diffusion to next-scale autoregression in the field of generative NVS, which avoids diffusion's "original sins" in speed and scalability that stem from the need for repeated denoising steps. Next-scale autoregression is also a rapidly evolving technique, and many advances in this line of research can possibly be used as plug-in improvements to our method. Hence we believe ArchonView can function as a base model for future innovations that can further drive this paradigm shift.

While our work aims to mainly explore the possibility of a paradigm shift and thus focused on the more fundamental task of NVS, we believe that our methodology can be easily modified to suit other downstream tasks. A particularly hopeful direction for future research is applying our method to 3D generation. This can be done by adopting a multi-view paradigm, as is common in current diffusion-dominated literature, such that consistency can be emergently imposed, facilitating conversion into 3D representations such as NeRFs, Gaussian splats, and meshes.

## 6 CONCLUSION

We introduce the first method of zero-shot single-image object-centric NVS to be based on visual autoregression, using the next-scale autoregression paradigm. We show that it does not require fine-tuning on 2D-pretrained checkpoints, achieves state-of-the-art performance across several benchmark tasks, is several times faster in inference time compared to previous methods, and demonstrates scaling behavior with model size and dataset size. This demonstrates the oft-overlooked potential of autoregressive backbones and their advantages over diffusion for NVS tasks, and provides a base model for potential future work in this direction.

Our main conclusions on the theoretical implications are: autoregression as a backbone paradigm for generative NVS may be more suitable than diffusion (in terms of accuracy and speed); differing from previous consensus, priors from 2D generation may not be necessary for generative NVS; and that scaling laws are expected to hold for generative NVS models with suitable architecture, as was shown for our design. We hope future work can derive more meaningful insights from our reframing of the problem.

## ETHICS STATEMENT

Results presented by this work, given their visual generative nature, are prone to being exploited by malicious agents. We encourage responsible usage in accordance with relevant common guidelines. The "Direct Use" portion of the DALL-E Mini model card (OpenAI, 2022), shared by works such as Stable Diffusion (Rombach et al., 2022), applies well.

## REPRODUCIBILITY STATEMENT

The code repository we provide in the abstract provides everything one needs to reproduce our results to their exact values (all random components were seeded as well to ensure this). This includes all used checkpoints, the training/validation/evaluation sets (already rendered and paired with pre-computed CLIP embeddings for the reader's convenience), the training/inference code, and package requirements. **We strongly encourage readers to try out our code.** Note that while Anonymous GitHub sometimes shows "The requested file is not found," this does not seem to affect downloading the repository.

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

Table 3: **Specifications for the checkpoints of varying sizes we trained.**

| Depth | 12 | 16 | 20 | 24 | 30 |
|---|---|---|---|---|---|
| # of Parameters | 135M | 311M | 601M | 1.0B | 2.0B |
| Training Time (h) | 14.5 | 17.1 | 23.5 | 30.0 | 51.3 |
| Inference Time (ms) | 160 | 180 | 200 | 220 | 260 |

## A  IMPROVING NVS BENCHMARKING

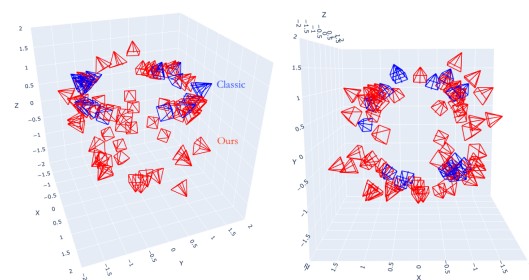

We notice that benchmarking in previous works have usually inherited the evaluation rendering settings used by Zero 1-to-3 (Liu et al., 2023). However, in this "classic" setting, we notice that input views are selected from a narrow range of elevation angles, which makes the task much easier but does not demonstrate the model's robustness across a wide range of possible views of an object. This is also a discrepancy with the training setting, where camera poses are sampled across an entire viewing sphere (save for very high elevation angles, and with the radial distance varying).

Figure 10: **Improving NVS evaluation.** A free view and a top view of the "classic" input camera views compared to ours (40 each sampled).

Hence, we choose to make the evaluation setting exactly the same as the training setting. A visual comparison between cameras sampled in our evaluation setting and the "classic" one is in Fig. 10. We note that this makes the task considerably more difficult (but the camera poses are still all within the training range, and hence are not out-of-distribution), and requires models to be more robust with respect to the input pose.

Regarding more details on the evaluation dataset, for all datasets except NeRF, we sample 100 objects randomly, and render 7 views for 3D objects (for RTMV we directly sample 7 views), with one being used as the input view and the other 6 being used for evaluation; for NeRF, we use all 8 objects, sample one random input view from each object's training set, and evaluate across all 200 testing views.

## B  TRAINING SPECIFICATIONS

Optimization of our model is conducted via AdamW (Loshchilov & Hutter, 2019), and implementation uses PyTorch (Paszke et al., 2019). We trained on 32 NVIDIA H200 GPUs across four nodes on a server cluster, and tested inference speed on a single H200. We report baseline results from officially recommended default settings.

We trained each of our released checkpoint models for 100 epochs on the full Objaverse dataset (800k objects), with a base learning rate of $8 \times 10^{-5}$ (result of grid search). We randomly render 12 views at 256×256 resolution from each object, and randomly select two as the input and target during each iteration. Information regarding training time and numbers of parameters are found in Tab. 3. During rendering, aligning with the common approach as in works like Liu et al. (2023), we randomly sample on a sphere with radius uniformly chosen from $[1.5, 2.2]$, and use the same setting for evaluation (as shown in Fig. 10).

## C  BACKBONE ARCHITECTURE DETAILS

**Multi-Scale Tokenization**   The VQVAE contains an image encoder-decoder pair $(\mathcal{E}, \mathcal{D})$, a quantizer $\mathcal{Q}$, and a learnable codebook $Z$ with vocabulary size $V$. The image encoder takes an image $x$ as input and encodes it into a feature map $f = \mathcal{E}(x) \in \mathbb{R}^{h \times w \times C}$ (where $h,w$ are the latent height/width, and $C$ is the embedding dimension). Note that the latent dimensions of the image are based on patching, similar to conventional autoregression and latent diffusion. The feature maps are then quantized as $q = \mathcal{Q}(f) \in [1..V]^{h \times w}$. The quantizer does this by mapping each patch $f_{i,j}$ to the Euclidean nearest codebook entry $Z_v$:

$$q_{i,j} = \arg\min_{v \in [1..V]} ||Z_v - f_{i,j}||_2. \tag{4}$$

The key to the multi-scale formulation is that the input image is resized into different resolution scales. For instance, since we use $16 \times 16$ patches, the $3 \times 3$ scale would reshape the image to $48 \times 48$ pixel resolution before tokenization; and the corresponding scale token would contain 9 patch tokens. All scales share the same codebook $Z$, thus ensuring a consistent vocabulary is used across different token scales. The image can then be reconstructed using the decoder $\mathcal{D}$ given the quantized tokens.

**Next-Scale Prediction Transformers**    Based on a frozen VQVAE the autoregressive model is formulated as follows. We attempt to achieve a probabilistic model $p_\theta$ (the next-scale transformer) conditioned by our inputs $(x, R, T)$ such that the joint likelihood of scale tokens $(r_1, r_2, \cdots, r_K)$ representing the distribution of $x_{R,T}^*$ can be modeled as

$$g(x, R, T) = p_\theta(r_1, r_2, \cdots, r_K | x, R, T) = \prod_{k=1}^{K} p_\theta(r_k | x, R, T, r_1, r_2, \cdots, r_{k-1}). \qquad (5)$$

Note that here the autoregressive assumption is that each scale $r_k$ only depends on the conditioning and the previous scales (similar to a coarse-to-fine process), instead of the later scales. We add conditioning tokens $c_k$ (as described in Sec. 3.4) as well for additional information.

We use teacher forcing to train the model, where $(r_1, r_2, \cdots, r_{K-1})$ of $x^*$ and conditioning from $x$ are used for causally predicting all tokens $(r_1, r_2, \cdots, r_K)$ of $x^*$. After sending inputs and conditioning through several transformer blocks, the eventual results are passed through a classifier head (consisting of a single linear layer) and outputted as classification logits, after which the cross-entropy loss between the logits and ground truths is taken for backpropagation.

Implementation-wise, each scale token $r_k$ is first interpolated to the same size as $r_{k+1}$ by taking the corresponding feature map $f$, resizing it accordingly, and applying tokenization. An exception is the $[s] \rightarrow r_1$ mapping, which does not require resizing. The upscaled results $e_k$ are then taken as input to the causal transformer. Each attention layer uses adaptive layer normalization (AdaLN) (Peebles & Xie, 2023) conditioned by the start token (to ensure consistency with the global encoding conditioning) and multi-head self-attention (Vaswani et al., 2017). We refer to the number of AdaLN transformer blocks per prediction step as the model depth and the dimension of tokens $C$ as the model width. For easy scaling, in our experiments we set the model width to always be 64 times the depth.

During inference, we first find $c_k$ and the start token $[s]$ using $x$. We then use them to infer the distribution of $r_1$ and sample from the distribution. We then use the available information to infer $r_2$, and so on. Note that already known tokens are kv-cached (Pope et al., 2023) and not replaced in further inference steps. The final inferred $r_k$ values are passed through the VQVAE decoder to arrive at $\hat{x}^*$.

