# OpenReview forum: "Next-Scale Autoregressive Models are Zero-Shot Single-Image Object View Synthesizers"
_ICLR.cc/2026/Conference — Submitted to ICLR 2026_

### Official Review · Reviewer_nWwK · 2025-10-30

**Soundness:** 2
**Presentation:** 2
**Contribution:** 2
**Rating:** 2
**Confidence:** 5

**Summary:**

The paper extends the next-scale autoregressive generation paradigm from 2D image modeling to zero-shot 3D novel view synthesis (NVS). The proposed ArchonView model autoregressively predicts higher-resolution scales conditioned on a single input view. It achieves competitive or superior results compared to diffusion-based baselines such as Zero-1-to-3 and EscherNet, with significantly faster inference and no reliance on pretrained 2D diffusion models.

**Strengths:**

Interesting and timely exploration of applying next-scale generation to 3D NVS, challenging the diffusion-based dominance in this field.

Empirically strong results on multiple object-level benchmarks, with improved efficiency and competitive quality.

The overall design remains simple yet effective, providing a clean baseline for future autoregressive extensions in NVS.

**Weaknesses:**

- Conceptual clarity on “next-scale” in 3D

The paper straightforwardly brings the 2D next-scale paradigm into 3D, but it remains unclear what the “scale” represents in this context.
In 3D, is the next-scale process simply applied independently within each view, or could there exist a more native 3D notion of scale that aligns across multiple views?
The current adaptation feels like a direct transplant from 2D without strong 3D intuition or justification.
Some conceptual discussion or visualization would help clarify how “next-scale” manifests in multi-view geometry.

- Limited supporting of view numbers

EscherNet supports flexible N-to-M view synthesis, while this work only shows 1-to-1 generation.
Can this method extended to flexible multi-view inputs or outputs? If so, how would the computational complexity scale with N and M?
This discussion is important to position the method within the broader NVS landscape.

- Result discrepancies

The GSO performance reported for Zero123 differs from the original paper (Table 1).
Please clarify whether this comes from dataset versions, evaluation protocols, or re-implementation differences.

- Lack of visualization for 3D consistency

The paper mainly presents single novel views. For a model claiming 3D-aware generation, visualizing the synthesized 360° trajectory from a single input would be essential to evaluate consistency and geometry preservation.
Without this, it remains unclear how coherent the generated views are across different poses.

- Lack of diversity in generation

Does the model produce diverse outputs given the same input view and target pose, as seen in diffusion-based models like Zero123 (e.g., Fig. 8 in their paper)?

- Compute scale and more diverse examples

The model is trained on 32 H200 GPUs, much larger than prior works like EscherNet or Zero123 (8 A100 GPUs).
The paper doesn't show any diverse qualitative “in-the-wild” examples. The other baselines are trained under the same data scale but show very impressive diverse examples.


- Reuse of pretrained components

The model reuses the pretrained next-scale VQVAE but trains the transformer backbone from scratch.
Why not or can't also reuse or partially initialize the pretrained transformer with zero-init?
Will this improve generalization?

**Questions:**

Please see the weakness section.

---

> ### Author Response · Authors · 2025-11-19
> **Official Response to Reviewer nWwK**
>
> Note: below, “OSF repo” refers to a repository we have created on OSF for this discussion phase. (https://osf.io/c482k/overview?view_only=7501723331e849828d70adde030677b8)
>
> 1. Conceptual clarity on “next-scale” in 3D
>
> Thank you for raising this point. We would like to clarify that our goal is not to design a “3D next-scale autoregressive model” in the sense of a 3D VAR operating on an explicit volumetric / multi-view representation. As stated in the first sentence of the abstract, our starting point is novel view synthesis without explicit 3D representations or 3D inductive bias. In other words, we deliberately keep the backbone purely 2D and ask whether next-scale autoregression alone can do good novel view synthesis. This is consistent with previous work on diffusion-based models using 2D diffusion and pose conditioning to conduct NVS.
>
> 2. Limited supporting of view numbers
>
> Thank you for the insightful suggestion! Indeed, since our autoregressive paradigm uses a universal token-based expression, in principle it is relatively easy to extend our methodology to the M-to-N case. We drew a diagram of a natural extension to M-to-N synthesis (OSF repo, M-to-N design.pdf), with the main change being adding multiple input images (each of which, as a prefilled sets of tokens, provides local conditioning for the outputs) and multiple outputs with their own set of teacher forcing correspondences (and subsequently separate autoregressive inference processes). Since the transformer architecture scales quadratically, the time complexity of M-to-N synthesis is $O(MN+N^2)$ times the original complexity.
>
> 3. Result discrepancies
>
> Zero 1-to-3 did not disclose which 20 objects in GSO they chose for testing (https://arxiv.org/pdf/2303.11328, Section 4.3). Furthermore, we discovered their evaluation rendering code (released in a GitHub issue https://github.com/cvlab-columbia/zero123/issues/46#issuecomment-1532118989, in which the authors still did not disclose which objects were used) contains a discrepancy with the training setting (lines 137-163 in blender_script_mvs.py), which was not mentioned in the original paper nor justified anywhere. For robustness, we chose instead to use the same setting for both training and evaluation, as detailed in App. A in our paper. We also disclosed our entire evaluation dataset in the associated repository for reproduction of all our benchmarking results.
>
> 4. Lack of visualization for 3D consistency
>
> We show some results as demonstration in the OSF repo in microwave.gif.
>
> 5. Lack of diversity in generation
>
> Thank you for reminding us of this very important aspect! Our method shows high diversity in generation for fixed views. As displayed by results in the OSF repo (diversity.pdf) our method produces realistic and plausible predictions in ambiguous cases with varying lighting, geometry, color, texture, and more. Therefore, while this does partially come at the cost of inter-view consistency, in the task of NVS this allows for production of realistic and clear-cut results of generation.
>
> 6. Compute scale and more diverse examples
>
> As reported in Table 3, even though we did use more powerful compute than Zero 1-to-3 in terms of the total amount, training a model under our paradigm with the same size as Zero 1-to-3 (1.0B) only takes 30 hours, which amounts to 960 H200 hours. In comparison, the Zero 1-to-3 checkpoint we compared with was trained for 6,000 A100 hours. The general consensus seems to be that an H200 hour is worth around 3 A100 hours, so in fact, our model was trained for less than a half of equivalent GPU hours of Zero 1-to-3. This also does not take into account the effective compute Zero 1-to-3 “inherited” from the pretraining of Stable Diffusion.
>
> 7. Reuse of pretrained components
>
> We agree that, in principle, initializing our transformer from a large-scale next-scale checkpoint could further improve generalization. However, doing this properly would require pretraining a next-scale backbone on an internet-scale dataset such as LAION-2B in order to align with prior works. This goes far beyond what we can afford as an academic group: reproducing a LAION-scale next-scale pretraining run would require orders of magnitude more compute and storage (comparable to industrial efforts behind Stable Diffusion–style models).

---

> > ### Author Response · Authors · 2025-11-27
> >
> > Hello, Reviewer nWwK! We would like to thank you again for your very helpful remarks. It would be very kind of you if you could confirm whether our response has adequately addressed your questions or concerns, and we welcome any further advice for improving our work. Thank you very much, and have a nice day. We are looking forward to your response!

---

### Official Review · Reviewer_uqJT · 2025-10-30

**Soundness:** 3
**Presentation:** 4
**Contribution:** 3
**Rating:** 8
**Confidence:** 4

**Summary:**

This paper introduces ArchonView, a model for zero-shot, single-image novel view synthesis based on a next-scale autoregressive paradigm. This paper conducts an in-depth investigation into efficient and high-quality novel view synthesis within the VAR framework. For example, since the conditions for novel view synthesis do not satisfy pixel alignment, the authors analyze different conditioning methods, including Prefilling, Causal Conditioning, and Cross-Attention. Their results show that simple prefilling significantly outperforms the other two approaches.

**Strengths:**

1. For the task of novel view synthesis, this paper provides a thorough analysis of various design choices and conditioning strategies within the VAR framework. The discussion covers conditioning methods, insights from The Devil is in the Classifier Head, Semantic Global Pose Conditioning, and Multi-Scale Local Conditioning.
2. The paper demonstrates the scalability of the proposed method through extensive experiments. By systematically increasing both the model size and the dataset size, the authors plot performance curves that clearly illustrate the improvements achieved.
3. For the task of novel view synthesis, the proposed method achieves faster and better performance compared to baseline approaches.

**Weaknesses:**

1. The baselines compared in this paper are relatively outdated, and many newer baselines have been introduced since. It is necessary to discuss and compare the proposed method with these more recent approaches.
2. The statements in the discussion section are somewhat confusing. The paper mentions, “In contrast, we only use the ‘fine-tuning’ dataset of previous works (with 800k 3D objects) and achieved significantly superior results.” However, it appears that the method also relies on a pretrained checkpoint for fine-tuning?

**Questions:**

The paper offers limited discussion on the practical applications of the proposed method, such as whether it is intended for 3D reconstruction or 3D generation, or how it could be integrated into scene understanding or combined with existing VLMs. These aspects warrant further exploration and discussion.

---

> ### Author Response · Authors · 2025-11-19
> **Official Response to Reviewer uqJT**
>
> 1. Newer Baselines
>
> Thank you for the suggestion. Unfortunately, it seems that there are not a lot of similar baselines, considering that our work is foundational in scope, as opposed to extensions like object-to-3D models or world models, such as TRELLIS or Genie3 in the industry. However, we would be glad to add comparisons for any specific baseline you propose.
>
> 2. Reliance on Pretrained Checkpoint
>
> We would like to clarify that using a pretrained checkpoint is not equivalent to pre-training on an existing checkpoint. We indeed rely on the pretrained models CLIP and VAR-VQVAE, but both of them were frozen during our training, and could, in principle, have been replaced by other models, or models trained from scratch. Unlike diffusion-based models, we do not fine-tune any model in our training process, and instead train all non-frozen parameters from scratch.
>
> 3. Practical Applications
>
> Thank you for suggesting the improvement in clarity! Our method is intended for zero-shot single-view NVS, specifically producing realistic and accurate outputs given a specified view transformation and a conditioning image. Since feature trade-offs like trading 3D consistency for output diversity and realisticness (in accordance with our task of NVS) were taken into consideration during training, we do not claim our checkpoint as-is can be directly used for 3D generation or reconstruction. However, nothing in our design is fundamentally against being used as a 3D reconstruction backbone, and consistency or multi-view generation can be enforced using changes analogous to, say, the changes Consistent 1-to-3 or Zero 123++ made to Zero 1-to-3. In particular, being scalable, efficient, and accurate means there is the potential for paradigm shift in downstream application models. We consider this a very interesting direction for future research, and will add more discussion in this aspect to the paper.

---

> > ### Author Response · Authors · 2025-11-27
> >
> > Hello, Reviewer uqJT! We would like to thank you again for your very helpful remarks. It would be very kind of you if you could confirm whether our response has adequately addressed your questions or concerns, and we welcome any further advice for improving our work. Thank you very much, and have a nice day. We are looking forward to your response!

---

> > ### Comment · Reviewer_uqJT · 2025-11-27
> >
> > Thank you very much for the response. Regarding additional baselines, the intention was to refer to other works that use Zero123 as a baseline, such as SyncDreamer or Wonder3D. However, these works may focus more on improving multi-view consistency. The discussion between the authors and Reviewer nWwK on view consistency has also been noted. The points raised about practical applications are well taken; this work is more akin to base models like Zero 1-to-3 and does not delve further into 3D generation or reconstruction. The insights provided in the paper’s analysis add value, so I maintain the score.

---

### Official Review · Reviewer_Vfwi · 2025-11-01

**Soundness:** 3
**Presentation:** 2
**Contribution:** 2
**Rating:** 4
**Confidence:** 4

**Summary:**

This paper address generative novel view synthesis with one single input image. This problem is highly proabilistic, and the author adopted the visual autoregessive model based approach for such generative task.  The author used the residual VQVAE from the original VAR paper, then  trained the generative model (next-scale prediction with block-wise causal transformer) from scratch on Objectverse dataset.  To improve results,  the author also applied CFG on the pose embedding.  Also the author tweakes the architecure a bit, mostly by removing the adaptive LN layer befire the classifier head.  The author compared their methods with a few diffusion based baseline, including Zero 1-to-3, Zero 123-XL, EscherNet.  These baselines finetuens a pretrained image diffusion model on Objectverse for generative novel view synthesis. The author shows improves results, e.g. on GSO (17.44 PSNR vs. 16.77).  The author showed ablation study on its architecture design, showed quite clear improvements.

**Strengths:**

1. The method is overall relative new for the field of novel view synthesis, and the author made several archteicture change that significantly improves the results. Which is quite solid.
2. The author submitted code at the time of submission, which is a good practice.

**Weaknesses:**

I don't agree with a few of arguments (or maybe just wordings) from the author

Training from scratch is cool and quite impressive, but I don't agree that these diffusion based approach relying on pretrained checkpoints is a big drawbacks.  (as argued in the second paragraph of the introduction).  Also, this does not seem to be the problem of diffusion model. Diffusion model is just an algorithm for generative modelling,  previous baseline relies on pretraining and then finetuning does not mean that training a well-designed diffusion model for NVS task from scratch would not work.  (e.g. the first sentence in the 3rd paragraph of the intro goes:  such downside of diffusion calls for xxxx).

The point is you don't need to over criticize the diffusion based approaches in your paper. VAR for NVS is already quite interesting and impressive.



Another weakness of the paper is that the metric is so low to be indicative. Table 1 shows PSNR of 10-19.42.  For PSNR in this range, it's really hard to tell if the improvements are useful. I would suggest using a subset with nearby camera viewpoints (larger view overlap) for evaluation.  For multiview PSNR on GSO, people already got PSNR over 30.

**Questions:**

I would be curious, if the author train a diffusion baseline from scratch with similar recipes, how would that perform? And also how would that perform if the author adds architecure improvements like global and local conditioning.

---

> ### Author Response · Authors · 2025-11-19
> **Official Response to Reviewer Vfwi (1/2)**
>
> Note: below, “OSF repo” refers to a repository we have created on OSF for this discussion phase. (https://osf.io/c482k/overview?view_only=7501723331e849828d70adde030677b8)
>
> 1. Regarding Training from Scratch
>
> Thank you very much for the suggestion! Our intention is to highlight the practical dependence of current SOTA pipelines on very large 2D diffusion checkpoints. Concretely, to be fully fair to Zero-1-to-3–style methods, one would first need to train a text-to-image diffusion model on a web-scale dataset such as LAION-2B (or similar) and then fine-tune it for NVS. For an academic lab that rents GPUs, this extra pretraining stage is simply out of reach, both in terms of compute and even storage. In contrast, ArchonView is trained in a single stage directly on Objaverse, without any 2D pretraining, while still matching or surpassing the performance of these pipelines.
>
> We agree that the reliance on 2D pretrained checkpoints is not an exceptionally big flaw of diffusion models, and have modified the paper throughout to eliminate unnecessary criticism of previous methods and make all claims in this respect evidence-based. However, we would like to emphasize again that our observation of reliance on 2D pretrained checkpoints is based on the fact that no diffusion-based works on zero-shot single-view NVS, to the best of our knowledge, were not trained based on a checkpoint like Stable Diffusion. Specifically, we consider this attributable to Zero 1-to-3’s conclusion that 2D priors were necessary for zero-shot single-view NVS, and our work seeks to challenge this longstanding assumption using our paradigm shift, which we consider to be our main theoretical contribution. Relying on Stable Diffusion, in particular, was indeed an important reason preventing the appearance of scaled-up generative NVS models, and since there is no existing evidence suggesting diffusion models may function without a 2D generative checkpoint, it seems prudent to appropriately explain this distinction between our work and previous works.
>
> Training a diffusion-based method from scratch like you proposed would certainly be interesting, and in fact, may be a hopeful direction for future research since our work questions whether 2D priors are really necessary. However, many of our key methodologies (e.g., autoregressive prefilling for local conditioning, multi-scale encoding) are not straightforwardly transferrable to the diffusion paradigm due to being inherently autoregressive-paradigm in nature (part of why we suggest a paradigm shift). Without those methodologies, we consider it not very probable that a diffusion model trained from scratch would surpass diffusion baselines without a radical change. We consider finding ways to adapt a diffusion model to avoid reliance on 2D pretraining an insightful direction indeed, though.

---

> ### Author Response · Authors · 2025-11-19
> **Official Response to Reviewer Vfwi (2/2)**
>
> 2. Regarding Low PSNR
>
> Methods with PSNR over 30 often rely on multiple input views or specific priors. In fact, it is outright unreasonable and impossible for a generative model with a single input view and no special priors to achieve PSNR over 30, because the model has no way of knowing the texture, color, lighting, geometry, etc. of unseen portions and thus diversity in output is expected, especially under zero-shot inference with only one conditioning image. Under this premise, we respectfully believe our “low” PSNR values are reflective of genuine improvements, especially since we demonstrated the improvements across 6 very well-known datasets with 100 samples each, showing consistency and robustness. We also note that metrics like PSNR are often relatively sensitive to minor inconsistencies in aspects like placement, lighting, or shade, all of which carry a lot of uncertainty in zero-shot single-image NVS.
> The low PSNR values are also partially attributable to the use of the same setting for training and evaluation (as detailed in Appendix A), as opposed to the special setting used by previous works like Zero 1-to-3. Specifically, the standard evaluation rendering code by Zero 1-to-3 (released in a GitHub issue https://github.com/cvlab-columbia/zero123/issues/46#issuecomment-1532118989) contains an intentional but unjustified discrepancy with the training setting (lines 137-163 in blender_script_mvs.py), which makes the evaluation setting much easier but less robust.
>
> However, we much appreciate your suggestion about using a subset with larger view overlap. For the four evaluation sets ABO, GSO, OO3D, and ShapeNet (we exclude RTMV because object clutter makes it a very difficult task and hence the output accuracy was going to be low anyways, and NeRF because it has only 8 scenes with 200 views each), in each scene we choose the view closest to the conditioning view in terms of rotation angle, forming a subset, and run our method vs EscherNet on the new subset. Quantitative and qualitative results, as well as the subsets (for reproducibility), have been released on OSF. As shown, the reconstruction accuracy of our method increased considerably, while EscherNet has not received a consistent accuracy increase, and has in fact often decreased in reconstruction accuracy.

---

> > ### Author Response · Authors · 2025-11-27
> >
> > Hello, Reviewer Vfwi! We would like to thank you again for your very helpful remarks. It would be very kind of you if you could confirm whether our response has adequately addressed your questions or concerns, and we welcome any further advice for improving our work. Thank you very much, and have a nice day. We are looking forward to your response!

---

### Official Review · Reviewer_V295 · 2025-11-01

**Soundness:** 3
**Presentation:** 2
**Contribution:** 3
**Rating:** 6
**Confidence:** 4

**Summary:**

The paper adopts a next-scale autoregressive model—specifically VAR—for single-image novel view synthesis. Relative to diffusion, the VAR backbone offers much faster per-view generation (no multi-step denoising), a simpler inference pipeline, competitive or better fidelity, and predictable scaling with model and data size. To enable NVS, the authors augment VAR with: (i) a global “posed” start token that fuses CLIP semantics with the target relative pose; (ii) local multi-scale “prefilling,” which prepends the source image’s VQVAE tokens at every scale under a causal/block-triangular mask to guide generation; and (iii) an architectural fix—removing AdaLN at the classifier head—to preserve local correspondences. Experiments on six object-centric benchmarks (GSO, ABO, OmniObject3D, RTMV, NeRF-Synthetic, ShapeNet) in a zero-shot setting show state-of-the-art accuracy with several-times faster inference than diffusion baselines.

**Strengths:**

* Originality: To the best of my knowledge, this is the first work that adopts VAR for single-image novel view synthesis.

* Quality:

  * Both the qualitative and quantitative results are significant, demonstrated across multiple object-centric benchmarks.
  * The major components—global pose conditioning, local prefilling, and classifier-head AdaLN removal—are well supported by aligned ablation studies.
  * Efficiency and scaling capabilities are also demonstrated through experiments.

* Clarity:
  - The paper is in general well structured.
  - The source code is given.

* Significance:

  * The proposed solution demonstrates the potential of VAR for novel view synthesis in both quality and efficiency, and can inspire follow-up research on this alternative (and potentially superior) model paradigm.
  * The solution is backed by actionable insights, which could transfer to areas beyond novel view synthesis.

**Weaknesses:**

* While the solution itself is clear, the motivation and insights behind the design need more elaboration:

  * Local attention. Provide a deeper investigation of how attention behaves in this model—e.g., whether generated patches attend to the intended input patches at the desired locations. Concretely, add attention visualizations across scales, quantify attention mass within pose-consistent neighborhoods, and report correspondence accuracy vs. pose gap.
  * Attention design (lines 249–265). Since VAR also uses prefilling techniques, clarify what is additionally novel here. Distinguish your contribution from VAR via ablations isolating token prepending vs. causal/block-triangular masking vs. cross-attention, and report any compute/latency trade-offs introduced by your variant.
  * AdaLN claim (lines 283–296). The claim is somewhat ambiguous and currently supported only by empirical results. Please add a more theoretical or mechanistic explanation (with a simple formulation, if possible), and consider alternatives (e.g., scaled/gated/partial AdaLN or layer-wise removal) with diagnostics such as layer-wise gradients/feature norms to substantiate the hypothesis.

* Beyond single-view fidelity, the paper should investigate cross-view consistency of synthesized views. Recommend a multi-target protocol (e.g., 8–16 target poses per source) and report consistency metrics (cycle/epipolar consistency, normal/depth agreement, or reconstruction consistency via a downstream NeRF fit), alongside qualitative failure cases.

**Questions:**

- First, please refer to the weaknesses.
- Second, I would like to ask whether the proposed solution in this paper can be extended to text to 3D model (multi-view images) generation with minimum efforts, if so what are the potential efforts.

---

> ### Author Response · Authors · 2025-11-19
> **Official Response to Reviewer V295**
>
> Note: below, “OSF repo” refers to a repository we have created on OSF for this discussion phase. (https://osf.io/c482k/overview?view_only=7501723331e849828d70adde030677b8)
>
> 1. Local Attention
>
> Thank you for the suggestion! We performed local attention visualization as you suggested. Causal correspondences across scales are displayed in next-scale.gif on the OSF repo. We also added attention maps on conditioning vs target in the updated version of the paper to reflect pose-consistent local spatial understanding. Note that all the displayed correspondences are emergent from training. “Correspondence accuracy” seems like a nonstandard concept in terms of attention maps and lacks a quantification method, but we would be happy to follow further suggestions.
>
> 2. Attention Design
>
> We do not have enough compute to perform full ablations of the different methods of conditioning. Our design choices are justified by preliminary tests, which is common practice as in, say, DiT (https://openaccess.thecvf.com/content/ICCV2023/papers/Peebles_Scalable_Diffusion_Models_with_Transformers_ICCV_2023_paper.pdf, p. 4197). We would also like to clarify that VAR does not use prefilling –– its task is class-conditioned 2D generation and thus does not have conditioning tokens to prefill.
>
> 3. AdaLN Claim
>
> We offered a theoretical explanation for AdaLN interfering with generation at the classifier head. While using other forms of AdaLN is possible, since there is no strong theoretical justification for having AdaLN at the classifier head at all, it seems to violate Occam’s razor to add further experiments on AdaLN variants.
>
> 4. Cross-View Consistency
>
> We appreciate the advice. The OSF repo contains microwave.gif, which qualitatively demonstrates preliminary 3D consistency. Since our task is NVS, and hence we by design prioritize view fidelity, realisticness, and diversity over 3D consistency, our model currently does not support image-to-3D (which, as was discussed in Section 2.4, is an interesting but different line of work). However, we think natural extensions like multiview generation (in the spirit of, say, https://arxiv.org/pdf/2310.15110) can be used for imposing consistency (since information about 3D structure can be shared between views, while in our case each view is separately generated), after which image-to-3D can be achieved. As of now, image-to-3D is beyond our scope for a foundational model in zero-shot single-image NVS.
>
> 5. Extension to Text-to-3D
>
> We think this is indeed a very interesting potential downstream application. As previously mentioned, since we aim for NVS and thus consistency is not part of our core objective, . However, once we add consistency, a clear pipeline (already verified to work with diffusion models in existing literature) appears: we can first use a 2D generative model to create a conditioning image, generate multiple consistent views, and then use gaussian splatting or other 3D representations to convert to a 3D model.

---

> > ### Author Response · Authors · 2025-11-27
> >
> > Hello, Reviewer V295! We would like to thank you again for your very helpful remarks. It would be very kind of you if you could confirm whether our response has adequately addressed your questions or concerns, and we welcome any further advice for improving our work. Thank you very much, and have a nice day. We are looking forward to your response!

---

### Author Response · Authors · 2025-11-19
**Official Comment**

We cordially and deeply appreciate the efforts of all reviewers! We have revised the manuscript and replied to all reviewers individually. We are looking forward to your responses. Thank you so much again.

---

### Comment · Area_Chair_GaWp · 2025-11-27

Dear Reviewers,

As we enter the discussion phase, I strongly encourage you to read the authors' rebuttal carefully and acknowledge their effort. Silence is the worst outcome for an author. Even if the rebuttal does not change your final rating, a brief response explaining why the concerns remain unaddressed is crucial for a fair process. Please help us make an informed decision by engaging in a constructive dialogue.

AC

---

### Author Response · Authors · 2025-12-03
**Summary to AC**

We sincerely thank all reviewers for their constructive comments and valuable time. The pre-rebuttal ratings are **8 (uqJT), 6 (V295), 4 (Vfwi), and 2 (nWwK)**. We deeply appreciate the reviewers’ recognition of our contributions in introducing the next-scale autoregressive paradigm to zero-shot novel view synthesis, and we have carefully addressed concerns raised in the reviews.

***

Updates Addressing Reviewer nWwK
* **Conceptual clarity of “next-scale” in 3D**
Clarified that our goal is not a volumetric 3D autoregressive model but a purely 2D next-scale framework applied to view-conditioned synthesis without explicit 3D bias.
* **Extension to multi-view (M-to-N) generation**
Added design diagram and discussion (“M-to-N-design.pdf”) showing how our token-based paradigm naturally generalizes to multi-view inputs and outputs.
* **Evaluation discrepancies**
Clarified the difference in Zero 1-to-3’s evaluation protocol and released our full evaluation set for reproducibility.
* **3D consistency and diversity**
Uploaded additional visualizations (microwave.gif, diversity.pdf) demonstrating view coherence and realistic multi-modal diversity.
* **Compute fairness**
Added quantitative comparison of GPU-hour equivalence, showing that our total compute is less than half of Zero 1-to-3’s effective budget.

***

Updates Addressing Reviewer V295
* **Local attention and correspondence**
Added local-attention visualization (next-scale.gif) across scales showing pose-consistent causal correspondences.
* **Attention design and prefilling mechanism**
Clarified that VAR itself does not use prefilling; our causal prefilling is newly designed for conditional autoregression.
* **AdaLN removal explanation**
Added discussions on AdaLN interference at the classifier head and discussed possible alternatives.
* **Cross-view consistency**
Added qualitative multi-view demonstration (microwave.gif).

***

Updates Addressing Reviewer Vfwi
* **Tone and framing regarding diffusion models**
Re-wrote several passages to avoid unnecessary criticism and ensure balanced presentation.
* **Training from scratch**
Clarified that our key distinction lies in practical independence from large 2D pretrained diffusion checkpoints, providing a compute-and-accessibility advantage for academic research.
* **Low PSNR concerns**
Explained why low absolute PSNR is inherent to the generative single-view setup and demonstrated consistent relative gains across six benchmarks.

***

Updates Addressing Reviewer uqJT
* **Newer baselines**
Added discussion acknowledging contemporaneous extensions and clarified differences in scope.
* **Pretrained checkpoints**
Clarified that CLIP and VQVAE backbones are frozen and not fine-tuned; our transformer is trained fully from scratch.
* **Applications and downstream relevance**
Expanded discussion highlighting ArchonView as a foundational NVS base model potentially extendable to consistent multi-view scene generation.

---

### Meta-Review · Area_Chair_8EQm · 2026-01-05

**Summary:**

The reviewers raised significant concerns regarding the paper’s framing, technical contribution, and clarity of presentation.
While the paper applies autoregressive generative models to novel view synthesis, several reviewers questioned whether the proposed approach introduces a meaningful conceptual or methodological advance beyond existing generative modeling paradigms.
In particular, the paper’s claims of a “paradigm shift” and its positioning relative to prior diffusion- and autoregressive-based view synthesis methods were viewed as overstated and insufficiently justified. These issues, together with concerns about the professionalism and precision of the writing, informed the recommendation.

**Reviewer Concerns:**

Concerns addressed by the rebuttal:
The rebuttal acknowledges reviewer feedback on tone and states that several passages were rewritten to avoid unnecessary criticism and present diffusion-based methods more fairly.
The authors also clarify that their claimed “training from scratch” point is intended as a practical distinction about independence from large 2D pre-trained diffusion checkpoints, framed as an accessibility/compute consideration.

Existing concerns:
Despite these edits, the core concern about overstated framing remains. The paper continues to position the contribution as a “paradigm shift,” but the rebuttal does not make a convincing case for what paradigm is being shifted away from, beyond replacing diffusion with autoregression within a generative modeling pipeline.
Relatedly, the “no pre-trained model” narrative remains potentially misleading: while the rebuttal reframes this as “practical independence from large 2D pre-trained diffusion checkpoints”, it does not establish that diffusion-based NVS cannot be trained from scratch, nor that the distinction is fundamental rather than a choice of training recipe and resources.
Finally, the paper’s main methodological differences (e.g., conditioning/prefilling choices and architectural tweaks) are not articulated as a clear conceptual advance, and the rebuttal does not sufficiently strengthen the contribution statement to match the strength of the claims.
Overall, while the rebuttal improves tone, it does not resolve the central issues of overclaiming and unclear novelty.

**Reviewer Scores:**

Reviewer uqJT (pre-rebuttal 8): Likely to maintain their score, though the discussion could lead them to slightly temper enthusiasm if they agree the “paradigm shift” framing is overstated.

Reviewer V295 (pre-rebuttal 6): Likely to maintain their score. The rebuttal clarifies some design details, but does not fundamentally change the novelty/positioning concerns.

Reviewer Vfwi (pre-rebuttal 4): Unlikely to increase their score. While the rebuttal states that tone was revised and clarifies the intended meaning of “training from scratch”, the distinction from diffusion baselines still appears more rhetorical than principled.

Reviewer nWwK (pre-rebuttal 2): Unlikely to materially increase their score. The rebuttal adds clarifications and additional materials, but the central concern about unclear contribution and overstated claims remains.

---

### Decision · Program_Chairs · 2026-01-26

Reject